# Prediction of mortality and prioritisation to tertiary care using the 'OUR-ARCad' risk score gleaned from the second wave of COVID-19 pandemic—A retrospective cohort study from South India

Narendran Gopalan[1]☯*, Vinod Kumar Viswanathan[2]☯, Vignes Anand Srinivasalu[1‡], Saranya Arumugam[1‡], Adhin Bhaskar[1‡], Tamizhselvan Manoharan[1], Santosh Kishor Chandrasekar[1], Divya Bujagaruban[1], Ramya Arumugham[1], Gopi Jagadeeswaran[1], Saravanan Madurai Pandian[2], Arunalatha Ponniah[2], Thirumaran Senguttuvan[3], Ponnuraja Chinnaiyan[1], Baskaran Dhanraj[1], Vineet Kumar Chadha[4], Balaji Purushotham[2‡], Manoj Vasanth Murhekar[3]

1 ICMR-NIRT-Indian Council of Medical Research -National Institute for Research in Tuberculosis, Chetpet, Chennai, India, 2 GSMC-Government Stanley Medical College and Hospital, Chennai, India, 3 NIE-Indian Council of Medical Research—National Institute of Epidemiology, Chennai, India, 4 NTI-National Tuberculosis Institute, Bangalore, India

☯ These authors contributed equally to this work.
‡ JW and HW also contributed equally to this work as second author and BP are **Senior author**
* gopalannaren@yahoo.co.in

## Abstract

### Background

Judicious utilisation of tertiary care facilities through appropriate risk stratification assumes priority, in a raging pandemic, of the nature of delta variant-predominated second wave of COVID-19 pandemic in India. Prioritisation of tertiary care, through a scientifically validated risk score, would maximise recovery without compromising individual safety, but importantly without straining the health system.

### Methods

De-identified data of COVID-19 confirmed patients admitted to a tertiary care hospital in South India, between April 1, 2021 and July 31, 2021, corresponding to the peak of COVID-19 second wave, were analysed after segregating into 'survivors' or 'non-survivors' to evaluate the risk factors for COVID-19 mortality at admission and formulate a risk score with easily obtainable but clinically relevant parameters for accurate patient triaging. The predictive ability was ascertained by the area under the receiver operator characteristics (AUROC) and the goodness of fit by the Hosmer-Lemeshow test and validated using the bootstrap method.

**Data Availability Statement:** All relevant data are within the manuscript and its Supporting Information files.

**Funding:** There was no funding available for this study.

**Competing interests:** The authors have declared that no competing interests exist.

**Abbreviations:** COVID-19, Coronavirus Disease-2019; AUC, Area Under the Curve; AUROC, Area Under the, Receiver Operating Characteristic; SaO$_2$, Peripheral Oxygen Saturation; NLR, Neutrophil-Lymphocyte Ratio; CRP, C-reactive protein; D-Dimer, Peak Dimerised Plasmin Fragment–D; CI, Confidence Interval; OR, Adjusted Odds Ratio; p, Probability; SE, Standard Error; vWF, von Willebrand Factor; CAD, Coronary Artery Diseas.

## Results

Of 617 COVID-19 patients (325 survivors, 292 non-survivors), treated as per prevailing national guidelines, with a slight male predilection (358/617 [58.0%]), fatalities in the age group above and below 50 years were (217/380 [57.1%]) and (75/237 [31.6%]), p<0.001. The relative distribution of the various parameters among survivors and non-survivors including self-reported comorbidities helped to derive the individual risk scores from parameters significant in the multivariable logistic regression. The **'OUR-ARCad'** risk score components were—**O**xygen saturation SaO$_2$<94%-**23**, **U**rea > 40mg/dL-**15,** Neutrophil/Lymphocytic ratio >3–**23**, **A**ge > 50 years-**8**, Pulse **R**ate >100–**8** and **C**oronary Artery disease-**15**. A summated score above 50, mandated tertiary care management (sensitivity-90%, specificity-75%; AUC-0.89), validated in 2000 bootstrap dataset.

## Conclusions

The OUR-ARCad risk score, could potentially maximize recovery in a raging COVID-19 pandemic, through prioritisation of tertiary care services, neither straining the health system nor compromising patient's safety, delivering and diverting care to those who needed the most.

## Introduction

As the danger of fresh waves of COVID-19 (Coronavirus disease 2019) pandemic kept lurking in the global horizon, a simplified memorisable risk score for the first wave was developed, segregating patients requiring tertiary care from those who could be safely retained at a primary care facility [1]. The delta variant predominating the second wave of COVID-19 in India imposed a greater demand for hospital beds, oxygen support and assisted ventilation, sparing neither the young nor those without comorbidities [2–5]. The immune fury, unleashed by the cytokine storm caused widespread tissue destruction, leading to organ failure, multisystem dysfunction, and ultimately death [6]. Advanced age and ischemic heart disease were important contributors for mortality in COVID-19 apart from laboratory parameters like absolute lymphocyte count, elevated Lactate Dehydrogenase and D-dimer levels [7]. Whereas some of the risk factors for mortality in influenza pneumonia resembled the prognostic indicators of COVID -19 like blood urea, absolute lymphocyte count and presence of cardiovascular disease, important differences also exist due to their unique pathophysiologies [8]. A brief overview of various studies aimed at deducing risk factors for survival across various levels of health system was already provided in our previous work on first wave of COVID-19 [1]. In this regard, we ventured to analyse determinants of survival from another tertiary care centre for COVID-19 in South India, using patient data collected retrospectively from medical records at admission, to generate a risk score, using easily elicitable clinical signs and simple laboratory parameters, readily available at a primary screening care facility. This stratified approach with appropriate referral to level of medical care based on their risk of mortality would prove extremely beneficial in future waves or pandemics, safeguarding the interests of patients while considerably reducing the health system burden, delivering the best medical care to the neediest.

## Methods

### Study design and setting

Retrospective case record review of patients with confirmed COVID-19 by real-time reverse transcription-polymerase chain reaction tests (obtained from either nasal or oropharyngeal swabs), admitted to a designated public sector COVID-19 tertiary care hospital between April 1, 2021 and July 31, 2021 were included in this analysis; the period roughly coinciding with the ascending slope of the second wave of COVID-19 in South India. Case records with near complete data were mined into a concise case record form, from August 18, 2022 until October 10, 2022, by five experienced clinicians, (GN, SA, SKC, RA and PST) after obtaining appropriate approvals. The de-identified data was further entered into an excel work sheet and checked for accuracy and completeness by two statisticians (CP, TM). Only the hospital admission number in the case record form formed the link. An independent statistician (AB) finally analysed the totally de-identified data. The case record forms were stored under lock and key by the Principal investigator (GN). Therapy and oxygen supplementation were as per prevailing national guidelines [9]. The study was approved by the Institutional Ethics committees, which accorded waiver from individual consenting due to the retrospective nature of the record review.

### Patient outcome categorisation

Patients were broadly categorised into two groups- viz 'survivors' (documented evidence of discharge with a normal oxygen saturation in room air at discharge) and 'non-survivors' (documented death during hospitalization).

### Variables considered for prediction

Demographic details describing age, gender, days to admission from the onset of symptoms and pre-existing comorbidities (self-reported) in the case sheets, along with vital signs first recorded upon admission, comprising of peripheral Oxygen saturation ($SaO_2$), pulse rate, and blood Pressure were transcribed into an excel sheet. Laboratory reports included plasma sugar (random), complete blood count focusing on neutrophil-lymphocyte ratio (NLR), serum electrolytes, liver, and renal function tests and inflammatory biomarkers wherever available. Baseline investigations of clinical relevance were also stratified based on clinical expertise and analysed both as categorical and continuous variables in order to deduce the risk scores.

### Statistical analysis

**Statistical methods applied.** Between 'survivors' and 'non-survivors', the relative distribution of all available clinically important and laboratory related relevant baseline variables were explored in the univariate analysis taking a 10% significance level. The chi-square test was used for proportions and Mann-Whitney U test for continuous variables based on their distribution, to explore the possibility of attribution of these factors to COVID-19 related mortality. Kaplan-Meier survival curves were constructed to provide a bird's eye view of the influence of each parameter on survival in COVID-19 disease. The backward stepwise multivariable logistic regression was used to find out the significant factors influencing mortality due to COVID-19 post-adjustment and those significant in the adjusted analysis were utilised to create the prediction model. Statistical analyses were performed using R software version 4.0.4 (R Core Team 2021 and IBM-SPSS Statistics applicable to Windows (IBM Corp. Released 2017, Version 25.0. Armonk, NY).

**Construction of the model and the risk score for the second wave.** The receiver operating characteristic (ROC) curves of those significant parameters in the multivariable logistic

regression were superimposed on the inflammatory biomarkers like C-reactive protein (CRP), peak dimerized plasmin fragment D (D-dimer) and ferritin to understand their relative accuracy in prediction. The Hosmer-Lemeshow test was done to establish the 'goodness of fit' of the model while the area under the receiver operating characteristic curve (AUROC), displaying the predictive capability, was ascertained using the receiver operator characteristics. The model was validated by fitting it into 2000 dataset generated using the bootstrap method [10]. Those parameters remaining significant in at least 70% of the bootstrap data, were selected for formulating the risk score which was calculated by multiplying the regression coefficient by ten and rounding off to the nearest integer [11]. The AUROC value was again estimated for evaluating the capacity of the risk score to predict mortality. The previous risk score acronymed 'OUR-ARDS' was also validated in this current dataset to compare its usefulness and relevance with respect to the second wave.

## Results

### Study population characteristics

Among de-identified case records of 617 COVID-19 confirmed patients, hospitalised during the peak of the COVID-19 second wave, there were 325 survivors and 292 non-survivors.

The COVID-19 second wave had a slight male predilection (358/617 [58.0%]). Fatalities in the age group above 50 years were comparatively more than the younger age group (217/380 [57.1%] vs 75/237 [31.6%], $p < 0.001$). The median days from onset of symptoms to admission in hospitals, was four days in each of the groups.

**Symptomatology, vital parameters and pre-existing comorbidities in the cohort.** Table 1 describes the relative distribution of the various parameters among survivors and non-survivors including self-reported comorbidities.

Among patients with saturation below <90 (on room air), 248/344 [72%] of them succumbed to the disease. On the other end of the spectrum, those who maintained an oxygen saturation >94 in room air, mortality was limited to (8/170 [4.7%]). There was at least one comorbidity self-reported by patients (347/617 [56.2%]), frequency being diabetes—(249/617 [40%]), hypertension—(176/617 [28.5%]) and coronary artery disease (40/617 [6.5%]). There were three cases of stroke, two with coexistent cardiovascular disease and one having mucormycosis with possible spread to the brain, presenting as both orbital swelling and stroke.

The details on severity, medication used, oxygen supplementation and complications that arose during hospitalisation are provided in S2 Table.

**Laboratory parameters and their accuracy with respect to biomarkers.** NLR>3, $SaO_2$ <94%, Urea >40 mg/dL, presence of coronary artery disease, age >50 years, hypertension, kidney disease and random blood sugar >200, were the clinically relevant parameters that emerged significant and further explored through the multivariable logistic regression to deduce the risk scores (Table 2).

In the second wave of COVID-19, the only comorbidity that retained significance post adjustment was self-reported coronary artery disease. The median levels of biomarkers, viz. D-dimer, CRP and ferritin in the blood, were strikingly higher among non-survivors than in those who survived (Table 1). Even though biomarkers were available only in a subset of patients, their AUC values were robust that compelled us to incorporate it in the ROC curves, to provide an illustrative comparison of the relative positions of the biomarkers with respect to the parameters that emerged significant post adjustment in predicting mortality in our study (Fig 2).

The D-dimer values at different levels of oxygen saturation in the cohort, along with their corresponding median NLR values are provided to show their interplay and influence on inflammation induced mortality (Fig 3).

**Table 1. Univariate analysis of baseline characteristics of hospitalised patients categorised into survivors and non-survivors**.**

| Variable | | Total (N = 617) n (%) | Survivors (N = 325) | Non-Survivors (N = 292) | OR (95% CI) | P-value |
|---|---|---|---|---|---|---|
| | | | n (%) | n (%) | | |
| Demographics | | | | | | |
| Age (Years)[#] | | 55 (42, 65) | 50 (39, 60) | 60 (48.25, 69) | 1.04 (1.03, 1.05) | <0.001 |
| Age (Years) | 50 and above | 380 (61.59) | 163 (50.15) | 217 (74.32) | 2.88 (2.04, 4.04) | <0.001 |
| Gender (Male) | | 358 (58.02) | 189 (58.15) | 169 (57.88) | 0.99 (0.72, 1.36) | 0.944 |
| Vital Signs | | | | | | |
| Pulse Rate (beats per min)[#] | | 94 (86, 108) | 90 (84, 102) | 100 (88, 110) | 1.03 (1.02, 1.04) | <0.001 |
| Pulse rate | >100 | 218 (36.5) | 83 (26.27) | 135 (47.87) | 2.58 (1.83, 3.63) | <0.001 |
| $SaO_2$ at Admission (%)[#] | | 89 (80, 95) | 95 (90, 97) | 80 (70, 88) | 0.85 (0.83, 0.88) | <0.001 |
| $SaO_2$ Level | < 90% | 344 (56.86) | 96 (30) | 248 (87.02) | 52.31 (24.76, 110.52) | <0.001 |
| | 90–94% | 91 (15.04) | 62 (19.38) | 29 (10.18) | 9.47 (4.11, 21.84) | |
| | ≥ 95% | 170 (28.1) | 162 (50.63) | 8 (2.81) | - | |
| Symptomatology** | | | | | | |
| Symptoms | | 587 (95.14) | 303 (93.23) | 284 (97.26) | 2.58 (1.13, 5.88) | 0.02 |
| Fever | | 396 (64.18) | 194 (59.69) | 202 (69.18) | 1.52 (1.09, 2.11) | 0.014 |
| Fatigue | | 40 (6.49) | 23 (7.1) | 17 (5.82) | 0.81 (0.42, 1.55) | 0.521 |
| Cough | | 402 (65.15) | 208 (64) | 194 (66.44) | 1.11 (0.8, 1.55) | 0.526 |
| Sore Throat | | 30 (4.86) | 24 (7.38) | 6 (2.05) | 0.26 (0.11, 0.65) | 0.002 |
| Head ache | | 33 (5.35) | 22 (6.77) | 11 (3.77) | 0.54 (0.26, 1.13) | 0.098 |
| Breathlessness | | 418 (67.75) | 168 (51.69) | 250 (85.62) | 5.56 (3.76, 8.24) | <0.001 |
| Chest Pain | | 10 (1.62) | 8 (2.46) | 2 (0.68) | 0.27 (0.06, 1.3) | 0.112 |
| Vomiting | | 21 (3.4) | 18 (5.54) | 3 (1.03) | 0.18 (0.05, 0.61) | 0.002 |
| Myalgia | | 101 (16.37) | 60 (18.46) | 41 (14.04) | 0.72 (0.47, 1.11) | 0.138 |
| Abdominal Pain | | 5 (0.81) | 2 (0.62) | 3 (1.03) | 1.68 (0.28, 10.1) | 0.569 |
| Altered Sensorium | | 13 (2.11) | 1 (0.31) | 12 (4.11) | 13.89 (1.79, 107.46) | 0.001 |
| Anosmia | | 21 (3.4) | 16 (4.92) | 5 (1.71) | 0.34 (0.12, 0.93) | 0.028 |
| Diarrhoea | | 32 (5.19) | 22 (6.77) | 10 (3.42) | 0.49 (0.23, 1.05) | 0.061 |
| Co-morbid Conditions | | | | | | |
| Co-morbidities | | 347 (56.61) | 153 (47.52) | 194 (66.67) | 2.21 (1.59, 3.07) | <0.001 |
| Diabetes Mellitus | | 249 (40.36) | 109 (33.54) | 140 (47.95) | 1.83 (1.32, 2.53) | <0.001 |
| Hypertension | | 176 (28.53) | 69 (21.23) | 107 (36.64) | 2.15 (1.5, 3.07) | <0.001 |
| Cardiovascular Disease | | 40 (6.48) | 8 (2.46) | 32 (10.96) | 4.88 (2.21, 10.77) | <0.001 |
| Kidney Disease | | 17 (2.76) | 4 (1.23) | 13 (4.45) | 3.74 (1.21, 11.6) | 0.015 |
| Others[ψ] | | 49 (7.99) | 12 (3.73) | 37 (12.71) | 3.76 (1.92, 7.37) | <0.001 |
| Laboratory Investigations expressed as median with interquartile range | | | | | | |
| RBS (mg/dL) | | 175 (113, 279.75) | 142 (100.5, 232) | 200 (137, 300) | 1.003 (1.001, 1.004) | <0.001 |
| RBS | >200 | 236 (42.22) | 102 (35.29) | 134 (49.63) | 1.81 (1.29, 2.54) | 0.001 |
| Urea* (mg/dL)[#] | | 37 (24, 57) | 27 (20, 38) | 52 (36.25, 73.5) | 1.06 (1.04, 1.07) | <0.001 |
| Urea | >40 | 255 (43.89) | 64 (21.26) | 191 (68.21) | 7.95 (5.47, 11.55) | <0.001 |
| Creatinine (mg/dL) | | 1 (0.8, 1.2) | 0.9 (0.8, 1.1) | 1.1 (0.9, 1.45) | 1.25 (1.06, 1.47) | <0.001 |
| NLR* | | 5.33 (2.76, 11.54) | 3.178 (1.72, 5.25) | 11.25 (6.65, 23.42) | 1.3 (1.23, 1.37) | <0.001 |
| NLR | 3> | 392 (72.46) | 156 (53.42) | 236 (94.78) | 15.83 (8.65, 28.95) | <0.001 |
| Albumin* (g/dL) | | 3.4 (3.1, 3.7) | 3.5 (3.3, 3.9) | 3.2 (3, 3.5) | 0.23 (0.15, 0.35) | <0.001 |
| Albumin | <3 | 67 (13.11) | 14 (5.22) | 53 (21.81) | 5.06 (2.73, 9.39) | <0.001 |
| CRP* (mg/dL) | | 60 (18, 91) | 22 (5.325, 63.925) | 84.3 (54, 98) | 1.01 (1.006, 1.014) | <0.001 |
| CRP | >20 | 296 (73.63) | 95 (50.53) | 201 (93.93) | 15.14 (8.06, 28.41) | <0.001 |

*(Continued)*

**Table 1.** (Continued)

| Variable | | Total (N = 617) n (%) | Survivors (N = 325) | Non-Survivors (N = 292) | OR (95% CI) | P-value |
|---|---|---|---|---|---|---|
| | | | n (%) | n (%) | | |
| LDH* (IU/L) | | 647(431,1037) | 481(373–646) | 1005(777–1900) | 1.003(1.002–1.004) | <0.001 |
| LDH | >500 | | 57(24.8%) | 93(40.4%) | 17.015 (7.33- 39.52) | <0.0001 |
| FERRITIN* (ng/mL) | | 597 (238, 1152.5) | 289 (121, 592.5) | 955 (488.5, 1486) | 1.002 (1.002, 1.003) | <0.001 |
| D-dimer* (ng/mL) | | 1060 (338.3, 3682.6) | 346 (200.35, 768.25) | 2554.2 (1060, 6170) | 1.001 (1.001, 1.001) | <0.001 |
| D-dimer | >750 | 31 (15.35) | 1 (0.78) | 30 (40.54) | 86.59 (11.47, 653.83) | <0.001 |

N–total number of participants, n, number; %, percentage; mg, milligram; dL, deciliter; RBS, random blood sugar; NLR, neutrophil Lymphocyte Ratio; g, gram; CRP, c-reactive protein; LDH–Lactate dehydrogenase; D-dimer–D protein dimer of fibrin degradation product; IU, International units; L, Litre; <, lesser than; >, greater than; ng, nanograms; ml, milliliter

#—median (Interquartile range)

*- data available N–number of values available from the sample and included in the analysis: albumin-511, Urea-581, NLR-541, CRP-402, Ferritin-318, D-dimer-407, LDH-230

**Data are presented as number (percentage) of patients unless otherwise indicated.

ψ- Detailed distribution of other comorbidities are provided in S1 Table.

**Model creation and the 'OUR-ARCad' risk score generation.** The Hosmer-Lemeshow test ($\chi^2$ = 8.43, p = 0.393) established the goodness of fit of the model with a concordance index (AUC) of 0.89 [95% CI, 0.86–0.92] that indicated an adequate discriminating power, excluding overfitting bias in the model. The sign of the regression coefficient was also found to be consistent across the bootstrap samples. The regression coefficient-based scoring system, the 'OUR-ARCad' score was derived from the sum of the individual scores assigned to each of the variables significant in the multivariable logistic regression model enlisted in Table 2. We elucidated that the sum of the risk scores with a cut-off of 50 and above served as the critical value for predicting mortality with a sensitivity of 90% and specificity of 75% and requires immediate escalation to tertiary care to avert fatality. The different ROC values (sensitivity and specificity) corresponding to the sum of the individual scores that constituted the net risk score for survival is listed in (S3 Table). The AUROC of the OUR-ARDs (1st wave risk score) validated in second wave dataset) and current OUR-ARCad score respectively were similar (0.88 vs 0.89) by serendipity (S1 Fig).

**Table 2. Multivariable logistic regression fitted to predict the factors affecting mortality among covid-19 infected patients admitted to hospital.**

| Variables | Multivariable Logistic regression | | | Coefficients' sign in bootstrap samples | | Coefficients' significance in bootstrap samples (%) | Risk score |
|---|---|---|---|---|---|---|---|
| | b (SE) | OR (95% CI) | p value | + (%) | - (%) | | |
| NLR (>3) | 2.27 (0.42) | 9.66 (4.22, 22.1) | <0.001 | 100.0 | 0.00 | 100.00 | 23 |
| SaO₂ (<94%) | 2.25 (0.35) | 9.44 (4.73, 18.87) | <0.001 | 100.0 | 0.00 | 100.00 | 23 |
| Urea (>40) | 1.45 (0.26) | 4.26 (2.54, 7.14) | <0.001 | 100.0 | 0.00 | 100.00 | 15 |
| CAD (Yes) | 1.51 (0.67) | 4.54 (1.23, 16.75) | 0.023 | 100.0 | 0.00 | 74.53 | 15 |
| Age (>50 years) | 0.77 (0.27) | 2.15 (1.26, 3.66) | 0.005 | 99.95 | 0.05 | 88.34 | 8 |
| Pulse Rate (>100) | 0.76 (0.27) | 2.15 (1.27, 3.64) | 0.004 | 100.0 | 0.00 | 87.57 | 8 |

NLR, neutrophil lymphocyte ratio; SaO₂, peripheral oxygen saturation; CAD—cardiovascular or coronary artery disease (self-reported); SE–standard error; OR–Odds ratio; %—percentage; CI–Confidence interval; +—plus'—- minus; <—lesser than. The parameters significant in the risk score formulation is included as Fig 1.

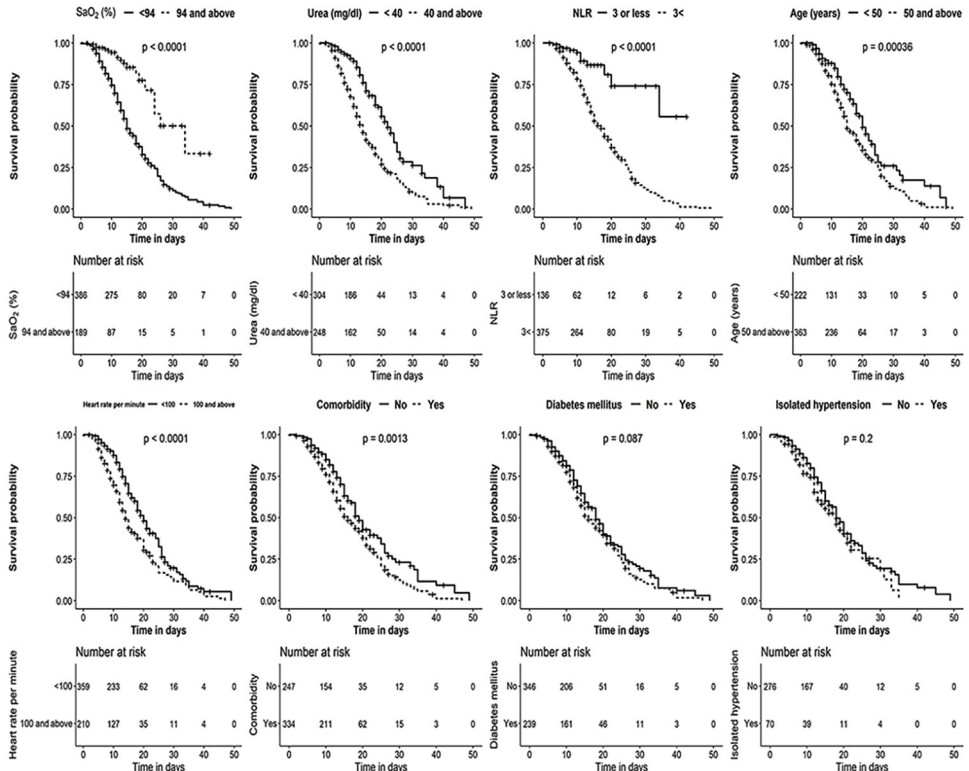

**Fig 1. Kaplan-Meier analysis of the parameters significantly attributing to risk of mortality in the univariate analysis.** Kaplan-Meier mean survival estimates for the time to mortality censored at 50 days. The numbers given below denote the number of individuals who were alive at that particular time point in days, with and without the risk factor that is being evaluated. Significance was computed using the log-rank test. $SaO_2$ –Peripheral oxygen saturation, %—percentage, <—lesser than, mg/dL–milligram per decilitre, NLR–Neutrophil Lymphocyte Ratio, p–p value.

## Discussion

In the rampaging second wave of COVID-19 pandemic predominated by the delta variant, the 'OUR-ARCad' risk score, built on simple parameters, easily obtainable even at a primary care centre (AUC of 0.89), was able to efficiently identify patients in need of early escalation to tertiary care, adequately differentiating them from those who could be safely managed at a primary health care or at home. This innovative strategy to quantify risk at an earlier period possessed the dual advantage of not only ensuring individual patient safety but importantly averting a huge strain on the health system, through evidence-based triaging [12]. This approach also reassures a patient to avert or avoid the stress of unnecessary admission, without any medical need. A cut-off value of the 'OUR-ARCad' score of 50 or more, achieved a sensitivity of 90% and a specificity of 75%, in predicting mortality and should alert the physician to escalate the patient's management to a tertiary care centre. We intentionally omitted the CT-SS score as the computed tomogram manifestations in the lungs were highly dependent on the stage of COVID-19 illness and misled the physician on severity [13]. We further recommend repeated measurements of the risk score every 3–5 days or alternatively estimate biomarkers if the scores fall in the border zone of 38–49 (S3 Table), in which case an increasing NLR provides a useful clue, as a harbinger of deterioration [14]. NLR and $SaO_2$ played a hegemonic role in COVID-19 survival. It is not surprising that the predominant symptom in the second wave among non-survivors was breathlessness. The hyper-inflammatory state reflected by the higher NLR, led to larger reductions in surface area of the respiratory membrane,

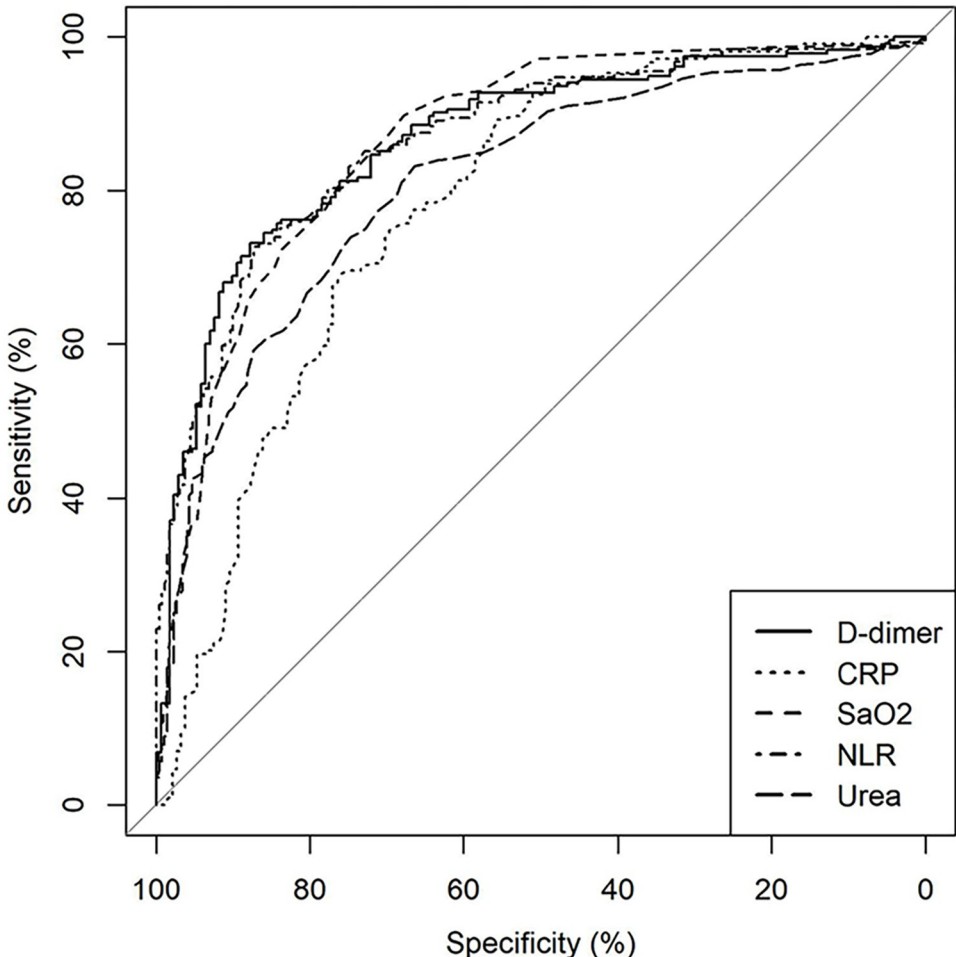

**Fig 2. Illustrated depiction of ROC curves of important parameters used in score formulation along with biomarkers.** The relative positions in the ROC curve also provide the precision of parameters, compared to biomarkers, in predicting mortality. ROC Curve—Receiver Operating Characteristic Curve, D-dimer–D protein dimer of Fibrin degradation product, CRP- C-Reactive Protein, $SaO_2$—Peripheral Oxygen Saturation, NLR—Neutrophil Lymphocyte Ratio, %—percentage.

resulting in a profound fall in $SaO_2$, as observed among non-survivors compared to survivors in our study (Table 1). Monitoring NLR, hence, serves as a close guide for predicting survival even while using steroids for prognostication in COVID-19 patients, unless complicated by inter-current infections [14].

The WHO guidelines 2020 and the ICMR national guidelines 2021 clearly emphasizes the need for maintaining a $SaO_2$ of at least 94% [9,15]. $SaO_2$ directly delivered the death penalty if saturation dipped below 90%, with the mortality doubling for every 5% drop in saturation thereafter [16]. The corresponding NLR value also proportionately increased in such cases [16,17]. The Liu et al. study demonstrated the disparity in $SaO_2$ between survivors and non-survivors, with nearly 75% of patients who succumbed, had a $SaO_2$ of <94% compared to 11% in those who survived, echoing our findings. Low oxygen saturation had the second highest Odds Ratio (OR) for mortality (5.80 [95% CI, 3.55–9.48]; P<0.001) after age [18]. The study conducted by Munoz et al. among the elderly showed that a peripheral oxygen saturation <92% had an OR of (5.85 [95% CI, 2.89–11.84]; P<0.001), independently predicting death

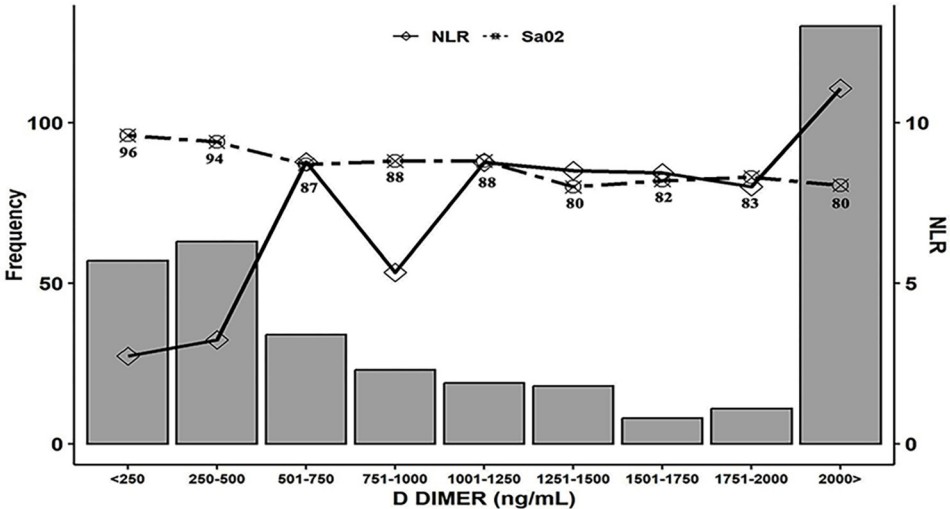

**Fig 3. Decreasing oxygen saturation with corresponding levels of NLR and D-dimer for understanding the interplay of inflammation.** The rising slope of NLR precedes the rise in D-dimer above 1000 ng/mL and corresponding reduction in $SaO_2$, making it a perfect surrogate for inflammatory biomarkers to dictate the prophecy of COVID-19 prognosis. The cut-off of $SaO_2$ above 94% as stated in the WHO and national guidelines is ascertained in our cohort when the D-dimer levels are well below 500 ng/mL (negative for in situ thrombosis). NLR–Neutrophil Lymphocyte Ratio, $SaO_2$ –Peripheral oxygen saturation, D-dimer–D protein dimer of Fibrin degradation product, ng/mL–nanogram per millilitre, <—lesser than, >—greater than.

[19]. Strangely, in our study, eight patients died despite maintaining an oxygen saturation of above 94. Further probing into their data showed that seven of the eight had diabetes, with four of them having a very high NLR >5 and D-dimer levels >2000 ng/mL signifying widespread inflammation though they could maintain their oxygen saturation.

Age above 50 years was another prominent predictor of survival, with proportion of younger patients succumbing to COVID-19 being (75/292 [26%]) and (60/259 [23%]) in the second and first wave respectively [1]. But studies globally have revealed a mixed scenario. While the second wave in Spain showed a higher median age, the study by Zirpe et al. however found no difference in the age groups, paralleling our findings [20,21].

An intact functioning kidney maintaining normal levels of blood urea is often a prerequisite for survival in COVID-19 [22]. The median value of blood urea doubled among those who succumbed compared to those who survived in our study (52 vs 27 mg/dL, p<0.001). Elevated urea levels facilitated transformation and splitting of the pentameric CRP to its active monomeric counterpart augmenting inflammation and tissue destruction, which in turn reduced the extracellular volume. This vicious cycle wound end fatally, unless broken by anti-inflammatory therapy and cautious fluid replacement [23]. Mudatsir et al. proposed that direct involvement of renal cells by the virus rather than due to pre-existing renal disease to be the cause [24]. We hypothesize that elevated Urea levels in the background of poor renal reserve as in chronic kidney disease would prove detrimental. Renal impairment not only delayed the clearance of cytokines, but elevated levels of von Willebrand factor (vWF), adding to both exaggerated inflammatory response and pro-coagulable state; further exacerbated by added vasodynamic insults incited by pre-existing diabetes or hypertension [25–27].

Self-reported coronary artery disease (CAD) was the only comorbidity significantly associated with mortality in our cohort when NLR and Urea were taken into the model. Cardiovascular disease whether recent or pre-existing did significantly contribute to fatality in patients, with accompanying high NLR worsening the prognosis [14,23,27]. We hypothesize that the

intense inflammation with steep increase in NLR, overshadowed the influence of other comorbidities like diabetes emerging statistically significant as an individual predictor of mortality. King et al. in their retrospective cohort of 13,000 COVID-19 patients, whose data were also collected at or around the time of admission, as in our study, demonstrated that vascular and coronary insufficiency were the sole comorbidities significantly influencing mortality, after adjustment for inflammatory parameters, akin to our findings [28]. That the spike protein, cleaved by plasmin, was abundant among CAD patients, could be one plausible reason for the increased severity and vulnerability of CAD patients to COVID-19 [29]. As diabetes indirectly influenced NLR, its direct effect would have been overshadowed by NLR in the adjusted analysis [30]. On the contrary, those studies which did not take NLR into the adjusted regression model had comorbidities as an independent determinant of prognosis [31]. With the other factors constituting the risk score remaining the same, the only factor that differed between the previous and current risk score was the comorbidity component alone [1]. Hence, the two risk scores inevitably had similar prediction potential; except that the OUR-ARCad score was more conservative with lesser admissions (S1 Fig). This was also supported by the findings of Kerai et al. that did not single out a specific comorbidity to influence survival greatly nor was differently distributed between the two waves [32]. Increase in pulse rate had been a consistent feature of severity in COVID-19 especially in the absence of fever and independently determined survival. The study by Abdel Ghaffar et al. showed an OR of (1.65 [95% CI, 1.22–2.23]; P<0.001), when the pulse rate exceeded 100 beats /min in alignment with our study [33].

We chose the most predatory Delta wave so that score derived from such a wave, would be optimistically useful in future pandemics as it could easily cover the variants of lesser severity and virulence, with more beds to spare in hospitals. The study by Kundavaram et al. had shown similar determinants of mortality, largely attributable to the delta variant that was prevailing at that time [34]. It was also noteworthy to find that those countries which had the second wave caused by the alpha or beta variant [as in Iran and Germany], instead of the delta variant, had similar or in fact improved outcomes in the second wave compared to the first wave of COVID-19 pandemic [35,36]. This approach paves the way for future exploration in other viral pandemics so that similar risk scores could be developed that would be useful for timely triaging but conserving resources and reducing health system burden at the same time. The main strength of the study is the formulation of a simple deducible risk score at admission, assessable at a peripheral health centre. The two scores were similar though data originated from two different tertiary care hospitals, in two different waves and probably strains. 'Silent Hypoxia' was also not ignored by this model. To get to the predictors, we used only data from documented survivors (discharged without oxygen supplementation maintaining their $SaO_2$) and documented hospital deaths. Clinical records confirming a transfer or readmission from another hospital to this tertiary care facility was a pre-specified exclusion. The predictors of COVID-19 mortality could also play a vital role in allied respiratory viral illness or future COVID-19 pandemics with suitable modifications of the risk score, where neutrophils and NLR play a major role in inflammation and tissue destruction [37–41]. The main limitation of the study is that it is primarily unicentric, catering to an adult population only. The selection of cases precludes calculation of case fatality rates. Variations in geographical location and strain of the virus should be considered while interpreting the results. With the second wave creating a mayhem, missing data were the rule and selection bias was possible as the analysed cohort came from the group that were fortunate to gain entry into the hospital, representing a population that were sicker and more moribund. There is a possibility to have left out patients who were asymptomatic or had minimal symptoms due to triaging. There could be patients who would have died even before reaching the hospital. But optimistically, we assume that their attributes may not be significantly different in clinical or lab parameters from those who

had been admitted and included and analysed in the study. D-dimer and CRP estimation should not be ignored where facilities exist. Some of the comorbidities were poorly represented in our cohort like stroke, malignancy, chronic renal disease, and airway disease. Hence, their potential role could not be assessed independently with accuracy. A more comprehensive risk score is possible with addition of D-dimer and CRP in future, but our intent was to triage at the time of entry into the hospital, where the inflammatory biomarkers may not be available at the primary health care facility. However, the parameters have been able to act as effective surrogates as shown in Fig 2. The illustrative comparison is precisely provided for this reason.

## Conclusions

The OUR-ARCad risk score, derived from easily obtainable parameters at a primary health care centre, offers a reliable triaging strategy to simplify COVID -19 patient care for future waves by ensuring admission for those who needed it while safely retaining the others in a primary health centre, allaying anxiety and effectively improving treatment outcomes in both the groups with the least burden on the health system.

## Supporting information

**S1 Fig. Comparison of the Area under the curve (AUC) values for the 'OUR ARCad' and OUR ARDS risk score (validated) with their corresponding sensitivity and specificity from second wave of COVID-19 in South India.** OUR-ARDs score can be found in reference 1 "Gopalan N, Senthil S, Prabakar NL et al. Predictors of mortality among hospitalized COVID-19 patients and risk score formulation for prioritizing tertiary care—An experience from South India. PLoS One [Internet]. 2022;17(2): e0263471. Available from: http://dx.doi.org/10.1371/journal.pone.0263471". OUR-ARDs and OUR-ARCad: O–peripheral oxygen saturation in percentage, U–urea in milligram per decilitre, R–Neutrophil lymphocyte ratio, A–age in years, R–Pulse rate in beats per minute, D–Diabetes mellitus, Cad–coronary artery disease / cardiovascular disease, %—percentage, AUC–Area under the Curve.
(TIF)

**S1 Table. Distribution of other comorbidities between survivors and non-survivors in detail (mentioned as "others" in Table 1).** Footnote: Numbers were too small to be individually analysed, but provided to have a bird's eye view of the population. COPD–Chronic Obstructive Pulmonary Disease, HIV–Human Immunodeficiency Virus.
(DOCX)

**S2 Table. Treatment details including specific antivirals, steroids and anticoagulants administered, oxygen supplementation and complications that arose in the cohort.** The higher percentage of patients using the drugs and injections simply implies that they had more severe disease mandating these interventions. This should not be interpreted that the drugs had actually caused the complications or death. N–total number of participants, %—percentage, O2 –oxygen, <—lesser than. [a]–Includes both low molecular weight heparin (LMWH) and unfractionated heparin (UFH).
(DOCX)

**S3 Table. Receiver operator curve values for the 'OUR ARCad' risk score with their corresponding sensitivity and specificity from second wave of COVID-19 in South India.** The 'OUR-ARCad' risk score of 50, calculated from the sum of the component scores if positive (Table 2) has a sensitivity of 90% and a specificity of 75%. This looks ideal when the referral to medical care is contemplated from a peripheral heath centre. However, when it is in border zone of 38 or more, but less than 50, re-scoring after 3–5 days or estimation of biomarkers if

facilities exist is recommended. OUR-ARCad—O–peripheral oxygen saturation in percentage, U–urea in milligram per decilitre, R–Neutrophil lymphocyte ratio, A–age in years, R–Pulse rate in beats per minute, D–Diabetes mellitus, Cad–coronary artery disease / cardiovascular disease, COVID-19 –Corona virus disease 2019.
(DOCX)

**S1 File. STROBE checklist.** The STROBE Checklist for this manuscript.
(PDF)

**S2 File. Participants anonymised data.** Anonymised data attached as an MS excel format.
(XLSX)

## Acknowledgments

We are grateful to the Ministry of Health & Family Welfare, Government of India; the Indian Council of Medical Research and the Department of Medicine & Family Welfare, Government of Tamil Nadu; for permitting the study and voluntarily offering the services of the facilities to be utilised for a successful publication. Our special thanks to the former and current Director Generals of ICMR & Secretaries -Department of Health Research, Government of India, Dr. Nivedita, Head, ECD division, ICMR, and Dr. J. Radhakrishnan IAS, Principal Secretary to the Government of Tamil Nadu, Dr.Shanthi Malar, DME, Dr.Sumathi Senthil Asst Prof, chengelpet Medical college, Dr.E.Prabhu, HOD nuclear medicine and Prof. (Dr.) Sridhar Rathinam MD, Vice-chancellor, Chettinad Medical college and research institute, for their guidance and mentorship throughout the study period. We thank our former HOD, Dr. Baskaran Dhanraj, and former Director, Dr. Padmapriyadarsini C for her valuable remarks and encouragement. We would like to appreciate the efforts of the medical record section of Department of Government Stanley medical college especially Mr. Thanigachalam Manickam and team who had helped us organise this exploration. Our thanks to Mr. Rajasekaran Subramanian who had helped us in record collection. Devayani Sundaramoorthy and Sindhu Asaithambi for secretarial assistance.

## Author Contributions

**Conceptualization:** Narendran Gopalan, Vinod Kumar Viswanathan.

**Data curation:** Narendran Gopalan, Vignes Anand Srinivasalu, Saranya Arumugam, Tamizhselvan Manoharan, Santosh Kishor Chandrasekar, Divya Bujagaruban, Ramya Arumugham, Gopi Jagadeeswaran, Thirumaran Senguttuvan, Ponnuraja Chinnaiyan.

**Formal analysis:** Narendran Gopalan, Adhin Bhaskar, Vineet Kumar Chadha.

**Investigation:** Narendran Gopalan, Vinod Kumar Viswanathan, Vignes Anand Srinivasalu, Saranya Arumugam, Adhin Bhaskar, Tamizhselvan Manoharan, Santosh Kishor Chandrasekar, Divya Bujagaruban, Ramya Arumugham, Gopi Jagadeeswaran, Saravanan Madurai Pandian, Arunalatha Ponniah, Thirumaran Senguttuvan, Baskaran Dhanraj.

**Methodology:** Narendran Gopalan, Vinod Kumar Viswanathan, Saranya Arumugam, Adhin Bhaskar, Tamizhselvan Manoharan, Thirumaran Senguttuvan, Ponnuraja Chinnaiyan.

**Project administration:** Narendran Gopalan, Vinod Kumar Viswanathan, Balaji Purushotham.

**Resources:** Saravanan Madurai Pandian, Arunalatha Ponniah, Baskaran Dhanraj.

**Software:** Adhin Bhaskar.

**Supervision:** Narendran Gopalan, Vinod Kumar Viswanathan, Balaji Purushotham.

**Validation:** Adhin Bhaskar, Tamizhselvan Manoharan, Ponnuraja Chinnaiyan, Vineet Kumar Chadha.

**Visualization:** Narendran Gopalan.

**Writing – original draft:** Narendran Gopalan, Vinod Kumar Viswanathan, Saranya Arumugam, Adhin Bhaskar, Manoj Vasanth Murhekar.

**Writing – review & editing:** Narendran Gopalan, Vinod Kumar Viswanathan, Saranya Arumugam, Adhin Bhaskar, Vineet Kumar Chadha, Balaji Purushotham, Manoj Vasanth Murhekar.

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
