## [Decision Letter · Decision Letter 0]

17 Sep 2024

PONE-D-23-23968Prediction of mortality and prioritisation to tertiary care using the ‘OUR-ARCad’ risk score gleaned from the second wave of COVID-19 pandemic - A retrospective cohort study from South IndiaPLOS ONE

Dear Dr. Gopalan,

Thank you for submitting your manuscript to PLOS ONE. After careful consideration, we feel that it has merit but does not fully meet PLOS ONE’s publication criteria as it currently stands. Therefore, we invite you to submit a revised version of the manuscript that addresses the points raised during the review process.

We look forward to receiving your revised manuscript.

Kind regards,

Siddharth Gosavi, MBBS, MD Internal Medicine,DNB Internal Medicine

Academic Editor

PLOS ONE

Additional Editor Comments:

Study is excellent.I highly recommedn your study Sir.Few observations must be made.You could have taken other blood ratios as well apart from NLR Ratio.Was Hba1C also recorded in your study population?Was LDH also included?How many of your patients also had stroke as a comorbidity?Definitely your study can serve as a marker for future where there will be an ease of burden on tertiary care hospitals.Introduction is too short.please elaborate further.Study is great.Please mention all the comorbidities in your study even the uncommon ones.

Also I must mention I received this article very recently for editing.The minimum number of reviews was not complete hence the delay.

Reviewers' comments:

Reviewer's Responses to Questions

**Comments to the Author**

1. Is the manuscript technically sound, and do the data support the conclusions?

Reviewer #1: Yes

2. Has the statistical analysis been performed appropriately and rigorously? 

Reviewer #1: Yes

3. Have the authors made all data underlying the findings in their manuscript fully available?

Reviewer #1: Yes

4. Is the manuscript presented in an intelligible fashion and written in standard English?

Reviewer #1: Yes

5. Review Comments to the Author

Reviewer #1: I appreciate this research work conducted by the authors and identifying the parameters to define the severity of COVID 19, which can be used at Primary Health Centre. The article is well written. Few of my observations are that since the COVID pandemic is almost over, will this risk score can be used for other Viral respiratory Illness like H1N1, etc.

Secondly, how did the authors decided about the sample size.

Thirdly, Authors could have also looked into the first wave data to see whether the same risk score also has similar result for the first wave, although the authors have compared the risk score of first wave with second wave.

Fourthly, This Specificity of only 75%. Hence I agree with the authors that more comprehensive score including inflammatory markers is required for better prediction.

6. PLOS authors have the option to publish the peer review history of their article (what does this mean?). If published, this will include your full peer review and any attached files.

Reviewer #1: No

---

## [Author Response · Author response to Decision Letter 0]

3 Oct 2024

Additional Editor Comments: 

Study is excellent. I highly recommend your study Sir. Few observations must be made. 

You could have taken other blood ratios as well apart from NLR Ratio.

Respected Sir, we wish to humbly admit that it was only a retrospective record review, hence only tests, which were performed in the government tertiary care hospital for COVID could be, analysed I this paper. Secondly, as we decided to use that parameter easily obtainable at a peripheral health centre, we felt we should go with basic parameters. NLR was available for almost all the patients and we could confidently analyse the data between survivors and non-survivors. We apologise that we do not have additional haematological parameters to analyse, even though a few had additional data.

Was Hba1C also recorded in your study population?

We appreciate your pertinent question sir. However, the hospital did not routinely perform HbA1c and only random blood glucose was available in the case records. 

How many of your patients also had stroke as a comorbidity?

Thank you for instigating us to explore into the data once again. There were three cases of stroke, one with CAD, the other one with CAD + Arrhythmia and the last one had mucormycosis with possible spread to the brain, presenting as both orbital swelling and stroke. Considering its importance as highlighted by the editor, This is now added in the text along with all details of “Other comorbidities” in the study provided separately as supplementary Table S3 . We request you to have it as a supplementary table for reducing the congestion in Table 1

Study is great. Please mention all the comorbidities in your study even the uncommon ones.

Sir, as advised by you, all the comorbidities are now included in supplementary table S3 

Was LDH also included?

LDH was done in the hospital for a limited number of cases. We had included it as advised by the esteemed editor in table 1 

In addition, I must mention I received this article very recently for editing. The minimum number of reviews was not complete hence the delay.

Our team of authors are indebted to you for the swift and dedicated review of this article. However, we were worried, as we had submitted it in the month of August 2023. We are extremely sorry if we had been little impatient.

.

Reviewers' comments: 

Review Comments to the Author

Reviewer #1: I appreciate this research work conducted by the authors and identifying the parameters to define the severity of COVID 19, which can be used at Primary Health Centre. The article is well written. Few of my observations are that since the COVID pandemic is almost over, will this risk score can be used for other Viral respiratory Illness like H1N1, etc.

This is an excellent question, as we did see that recently transient myocarditis had been reported in the vicinity of the third wave of COVID and these cases were not RT-PCR positive but were attributed to the Influenza epidemic occurring concurrently. But, we are optimistic that there could be translation to other viral epidemics as well but would require further exploration and appropriate collection of data for validation. This is mentioned in the text in page number 19 and referenced 35-39. It is an important point to ponder and we are grateful to the reviewer for this remark.

Secondly, how did the authors decided about the sample size.

Sir, this was a retrospective record review, aiming to evaluate risk factors for prediction of mortality and Triaging in the second wave of COVID-19, dominated by the Delta virus strain. Hence, we chose the peak of second wave of COVID-19, the most devastating one from April 1 2021 to July 2021, with data taken from near complete case sheets,. [Mentioned in methodology, page 7] Hence, the sample size was not determined Apriori. But gratefully, data is available for every parameter used in the model for logistic regression above 1:20 events per independent variable, sufficient for meaningful comparison. The significance deduced in the AUC curve and the bootstrap method has added further consistency and validity to the findings.

Thirdly, Authors could have also looked into the first wave data to see whether the same risk score also has similar result for the first wave, although the authors have compared the risk score of first wave with second wave.

This is a brainstorming question and we are grateful for asking this query. With the experience gained in the first wave and extra data available in the second wave, we could confidently infer that diabetes did not remain a significant factor for causing death when “cardiovascular diseases” was included in the model. As the most devastating wave was the second wave dominated by the Delta variant and this wave required more detailed triaging with a deluge of patients rushing into the hospital, throwing the health system into disarray, we had given more weightage to the second score that differed from the first score by just one component, diabetes replaced by Coronary artery disease. We had explained this in the text in the last para on page 17 and first para on page 18. This explanation has been added to the text for better clarity.

Fourthly, This Specificity of only 75%. Hence, I agree with the authors that more comprehensive score including inflammatory markers is required for better prediction.

We are grateful to the editor and reviewers for this comment and understanding clearly the limitations involved in a retrospective record review.

Data availability:

Anonymised data has been shared and uploaded as a supporting file to PLOS ONE, as requested. Thank you.

May I confirm the proposed data availability statement is okay to you? :"All relevant data are within the paper and its Supporting Information files." Once you confirm, we will update your Data Availability statement on your behalf using the information above.

Thank you for your email and for reviewing our submission. We have provided the de-identified and anonymised data as a supplementary (excel) file labelled as "S1_File_OUR_ARCad_Score_Anonymised_Data" along with the manuscript. Please include the statement "All relevant data are within the manuscript and its supporting information files." (replacing the word 'paper' with 'manuscript'). Please update the statement as required.

---

## [Editor Report · Decision Letter 1]

17 Oct 2024

Prediction of mortality and prioritisation to tertiary care using the ‘OUR-ARCad’ risk score gleaned from the second wave of COVID-19 pandemic - A retrospective cohort study from South India

PONE-D-23-23968R1

Dear Dr. Gopalan,

We’re pleased to inform you that your manuscript has been judged scientifically suitable for publication and will be formally accepted for publication once it meets all outstanding technical requirements.

Kind regards,

Siddharth Gosavi, MBBS, MD Internal Medicine,DNB Internal Medicine

Academic Editor

PLOS ONE

Additional Editor Comments (optional):

Thank you for the revised file.
---

## [Editor Report · Acceptance letter]

14 Nov 2024

PONE-D-23-23968R1 

PLOS ONE

Dear Dr. Gopalan, 

I'm pleased to inform you that your manuscript has been deemed suitable for publication in PLOS ONE. Congratulations! Your manuscript is now being handed over to our production team.

Kind regards, 

on behalf of

Dr. Siddharth Gosavi 

Academic Editor

PLOS ONE